# Commensal Bacterium *Rothia aeria* Degrades and Detoxifies Gluten via a Highly Effective Subtilisin Enzyme

**DOI:** 10.3390/nu12123724

**Published:** 2020-12-02

**Authors:** Guoxian Wei, Ghassan Darwish, Frank G. Oppenheim, Detlef Schuppan, Eva J. Helmerhorst

**Affiliations:** 1Department of Molecular and Cell Biology, Henry M. Goldman School of Dental Medicine, 700 Albany Street, Boston, MA 20118, USA; gdarwish@bu.edu (G.D.); fropp@bu.edu (F.G.O.); histatins@gmail.com (E.J.H.); 2Institute for Translational Immunology and Research Center for Immunotherapy (FZI), Johannes Gutenberg University (JGU) Medical Center, 5513 Mainz, Germany; 3Division of Gastroenterology, Beth Israel Deaconess Medical Center, Harvard Medical School, Boston, MA 02215, USA

**Keywords:** celiac disease, epitope, *Rothia*, *Bacillus*, gluten, degradation, subtilisin, commensal, detoxify, neutralize, cure

## Abstract

Celiac disease is characterized by a chronic immune-mediated inflammation of the small intestine, triggered by gluten contained in wheat, barley, and rye. *Rothia aeria*, a gram-positive natural colonizer of the oral cavity and the upper digestive tract is able to degrade and detoxify gluten in vitro. The objective of this study was to assess gluten-degrading activity of live and dead *R. aeria* bacteria in vitro, and to isolate the *R. aeria* gluten-degrading enzyme. Methods: After an overnight fast, Balb/c mouse were fed a 1 g pellet of standard chow containing 50% wheat (and 4% gliadin) with or without 1.6 × 10^7^ live *R. aeria* bacteria. After 2 h, in vivo gluten degradation was assessed in gastric contents by SDS-PAGE and immunoblotting, and immunogenic epitope neutralization was assessed with the R5 gliadin ELISA assay. *R. aeria* enzyme isolation and identification was accomplished by separating proteins in the bacterial cell homogenate by C18 chromatography followed by gliadin zymography and mass spectrometric analysis of excised bands. Results: In mice fed with *R. aeria*, gliadins and immunogenic epitopes were reduced by 20% and 33%, respectively, as compared to gluten digested in control mice. Killing of *R. aeria* bacteria in ethanol did not abolish enzyme activity associated with the bacteria. The gluten degrading enzyme was identified as BAV86562.1, here identified as a member of the subtilisin family. Conclusion: This study shows the potential of *R. aeria* to be used as a first probiotic for gluten digestion in vivo, either as live or dead bacteria, or, alternatively, for using the purified *R. aeria* enzyme, to benefit the gluten-intolerant patient population.

## 1. Introduction

Gluten is found in wheat, barley, and rye and triggers celiac diseases (CeD), a chronic inflammatory immune-mediated disease affecting the (proximal) small intestine [1]. Globally, CeD can be found in 0.5 up to 2% of wheat consuming populations [2]. Immunogenic gluten proteins show an unusual resistance to degradation by human digestive enzymes and are antigenically presented on human leukocyte antigen (HLA) DQ2 or DQ8 that are expressed on myeloid and B cells in the intestinal lamina propria, triggering a gluten peptide specific T helper 1 (Th1) T cell response [3,4]. Immunogenicity (toxicity) associated with gluten is peptide length-dependent, where peptides shorter than 9 residues are unable to generate an immune response [5,6,7]. Therefore, detoxification of gluten peptides can be achieved by their extensive proteolytic fragmentation to smaller gluten moieties.

The microbiome colonizing the oral cavity is a rich source of proteolytic enzymes. The oral cavity is also the port of entry where ingested food contents are mixed with saliva and reach the more downstream regions of the gastrointestinal tract. The oral microbiome comprises over 700 different microbial species or phenotypes [8,9]. Saliva harbors over 100 million bacterial cells per milliliter. In fact, the oral cavity is, after the colon and the cecum, the most densely colonized part of the human gastrointestinal tract [10]. The functions and properties of many individual species within these mixed culture microbiomes remain unknown. Furthermore, the potential contribution of, e.g., oral microbial species to physiological processes in the digestive tract is virtually unknown. 

We recently showed that oral *Rothia* bacteria produce enzymes that degrade dietary gluten proteins in vitro [3]. This finding of an oral (and gastric) microbe digesting an essential food constituent points to potentially novel functions for oral microorganisms in digestion, having implications beyond the oral cavity. The *Rothia* genus, formerly called *Stomatococcus*, comprises several species, including *R. aeria*, *R. mucilaginosa*, and *R. dentocariosa*. They are Gram-positive microbes that thrive well in the oral environment. *R. mucilaginosa* has also been detected at low levels in the duodenum [11]. Little is known on the functional role that *Rothia* species play in the oral cavity, other than that they are mostly associated with oral health [12]. Here, the discovery of the gluten-degrading capabilities of *Rothia* bacteria added to the potential beneficial activity of this commensal. Furthermore, the fact that *R. mucilaginosa* and *R. aeria* species [13,14,15] degrade gluten suggests they may be useful in the treatment of CeD, which is triggered by incompletely digested gluten peptides. 

The purpose of the present study was three-fold. First, to investigate the gluten degrading and epitope neutralizing activity of *R. aeria* in vivo; second, to assess if *R. aeria* bacteria would retain enzymatic activity after the bacteria are killed; and third, to isolate and identify the gluten-degrading enzyme from *R. aeria*.

## 2. Materials and Methods

### 2.1. In Vivo Digestion Experiment

Cultivation of *R. aeria* bacteria—*Rothia aeria* strain WSA-8 [15] has been deposited as strain HM-818 to the BEI resources (beiresources.org) and has been sequenced. It is equivalent to *R. aeria* Oral Taxon 188, strain F0474 (HMP ID 1324). *R. aeria* bacteria were grown on *Brucella* agar plates (Hardy Diagnostics, Santa Maria, CA, USA) at 37 °C for 48 h under aerobic conditions. The cells were harvested from the plate with a cotton swab and suspended in sterile PBS to an OD620 180. Aliquots of 1.5 mL of the suspension were centrifuged and the supernatants were removed. The cell aliquots comprising approximately 1.59 × 10^7^ cells/mL were lyophilized in a SpeedVac and stored at −80 °C. 

Preparation of mouse chow admixed with *Rothia* bacteria—Mouse chow (2018, Envigo, Cambridgeshire, UK) with bacteria was prepared in advance of the in vivo experiment. The chow used contains approximately 18% crude protein. An aliquot of 1 g of mouse chow, containing 50% wheat as protein source, containing roughly 10% of gluten and 8% gliadin, approximately 25 mg of gliadin was dry mixed with lyophilized *R. aeria* bacteria at a ratio of 1.6 × 10^7^ cells/gram chow. The mixture of *Rothia* bacteria and chow were added to a 2 mL Eppendorf tube and water was added incrementally in less than 1 min to form a slurry. The chow slurry was dried in the SpeedVac (SpeedVac plus SC110 A, Savant, NY, USA) for 3 h during which the slurry was frozen. The resultant pellet was removed from the Eppendorf tube and was visually similar to an original chow pellet. 

In vivo digestion by *Rothia* bacteria—Nine-week-old female Balb/c mice (n = 9 mice/group) were fasted for 24 h and then fed 1 g of mouse chow with and without added *R. aeria* bacteria. Mice in both groups ingested the chow within 1 h and then were allowed to digest the chow for an additional 2 h. Stomach and (proximal) small intestine were harvested. Stomach and small intestinal contents were collected and centrifuged (2000× *g*, ×10 min, Thermo Scientific legend X1R, USA). The supernatants were collected and boiled (100 °C, 10 min), then mixed with 60% ethanol to extract the gliadins by shaking (40 rpm) at room temperature for 1 h and centrifuged (2000× *g* × 10 min, 4 °C). The supernatants were collected, and protein concentrations determined with the BCA protein assay. 

Gliadin degradation—Since approximately one third of the food protein were found in the small intestine and two thirds in the stomach, and only negligible amounts of gliadin in chow were found in the small intestine (with less than 10% of the gliadin that were detectable in the stomach), gliadin degradation was only assessed in the harvested stomach samples by SDS PAGE and immunoblotting using a rabbit polyclonal anti-gliadin-peroxidase conjugated antibody produced (Sigma A1052-1ML). Densitometric analysis of the obtained signals was carried out by drawing equal size boxes around the bands in the gel of 37 kDa and 50 kDa bands, representing the α/β and γ gliadins, respectively. The data were corrected for intensity level after subtracting the background by selecting a blank area in the image on the different gels. The data from the various gels were normalized to 5 µg of gliadin (provider) loaded on each gel. Analyses of band intensities were carried out using Quantity one software. 

Gliadin epitope neutralization—Hydrolysis of gliadin immunogenic epitopes in the harvested stomach contents was assessed with the R5 ELISA assay (RIDASCREEN Gliadin-R-Biopharm AG #R7001, Darmstadt, Germany) that measures an immune dominant omega-gliadin epitope. The protein concentration of the stomach samples was adjusted to 160 µg/mL, and diluted 3200-fold in dilution buffer, securing linearity in relation to the standard curve. Aliquots of 100 µL of each of the diluted samples, as well as the controls of the 0 ppb and 80 ppb gliadin standard solutions included in the kit, were added to each well. The assay was conducted according to the manufacturer’s instructions.

### 2.2. In Vitro Gliadin Digestion by Live and Dead Rothia Bacteria 

*R. aeria* and *R. mucilagiosa* cell killing—The two oral isolates, *R. mucilaginosa* ATCC 25296 and *R. aeria* (strain WSA-8) were cultured on *Brucella* agar plates, harvested and diluted in triplicate to an OD620 of 1.2 in either 1 mL of 70% ethanol or in 1 mL of saliva ion buffer (SIB), containing 50 mM KCl, 1.5 mM potassium phosphate, 1 mM CaCl_2_ and 0.1 mM MgCl_2_, pH 7.0. The bacterial cell suspensions were incubated for 30 min at 37 °C and 5 µL aliquots were plated in triplicate on *Brucella* agar. The residual 995 µL aliquots were lyophilized to dryness using a SpeedVac (Thermo Fisher, Waltham, MA, USA). The lyophilized bacteria in 70% ethanol were reconstituted in 1 mL sterile SIB, and the lyophilized bacteria in SIB were reconstituted in deionized sterile water, to obtain the same final ion composition in both samples. The reconstituted cell samples were used to determine gliadin degrading enzyme activity.

In gel gliadin degradation—Zymogram gels (6%) were prepared using gliadin from wheat (Sigma, St. Louis, MO, USA) as the gel-incorporated substrate, as previously described [16]. In brief, bacteria from the above reconstituted suspensions equivalent to 600 µl, were harvested, suspended in 30 µL zymogram sample buffer, and loaded onto a zymogram gel. Electrophoresis was carried out at 100 V at 4 °C, and gels were renatured and developed in zymogram renaturing and developing buffers (InVitrogen, Carlsbad, CA, USA) according to the manufacturer’s instructions. After 48 h of development at 37 °C, gels were stained with 0.1% Coomassie Brilliant Blue as described [16].

Synthetic tripeptide substrates hydrolysis—In a 96 well microtiter plate aliquots of 200 µL aliquots of the reconstituted *Rothia aeria* and *R. mucilaginosa* cell suspensions were mixed with 20 mM Z-Tyr-Pro-Gln-pNA (Z-YPQ-pNA) (21st Century Biochemicals, Marlborough, MA, USA). The final substrate concentration was 200 µM. All experiments were performed in triplicate. Substrate hydrolysis was monitored spectrophotometrically at 405 nm, using a Genios microtiterplate reader (Tecan Group Ltd., Männedorf, Switzerland) and Deltasoft software, with the equipment temperature set at 37 °C. Readings were performed in the kinetic mode at selected time intervals to capture the linear part of the conversion curve. 

### 2.3. Isolation and Identification of the Gluten-Degrading Enzyme from R. aeria 

To identify the gluten-degrading enzyme produced by *R. aeria*, the bacteria were cultured on *Brucella* agar plates (Hardy Diagnostics, Santa Maria, CA, USA) at 37 °C for 48 h under aerobic conditions. The cells were harvested from the plate with a cotton swab and suspended in sterile PBS to an OD620 5.0. Four aliquots of 1.5 mL of the suspension were centrifuged (2000× *g*, 10 min, 4 °C) and the supernatants were removed.

SDS-PAGE and casein zymography—The four cell pellets were re-suspended each in 200 µL zymogram sample buffer [13], and subject to a non-reducing (no DTT or β-mercaptoethanol-containing) 6% SDS-PAGE gel of 16 × 20 × 0.15 cm, using a protean II xi cell system (Bio-Rad, Hercules, CA, USA). The composition of this gel was the same as that of previously published 6% gliadin zymogram gels but without incorporation of gliadin [13]. After electrophoresis at a constant voltage of 120 V at 4 °C, the gel was divided in two halves. The first half was stained with 0.1% Coomassie Brilliant Blue in 40% (*v*/*v*) methanol/10% (*v*/*v*) acetic acid. The second half was developed as a zymogram gel to reveal enzyme activity associated with the bands [14]. To this effect, the zymogram gel half was washed twice for 30 min in buffer containing 2.5% Triton X-100 (renaturing buffer; Life Technologies, Carlsbad, CA, USA), followed by washing twice for 1 h in buffer containing 20 mM Tris-HCl, pH 7.5 (developing buffer; Life Technologies, Carlsbad, CA, USA). Gel A was then incubated in developing buffer supplemented with 1% casein (Sigma, St. Louis, MO, USA) at 37 °C for 1.5 h, followed by a washing step for 2 min and staining with 0.1% (*w*/*v*) Coomassie Brilliant Blue in 40% (*v*/*v*) methanol/10% (*v*/*v*) acetic acid for 24 h. Both gel halves were then de-stained in 40% (*v*/*v*) methanol/10% (*v*/*v*) acetic acid until optimal contrast was achieved. 

LC-ESI-MS/MS—The gel halves described above were aligned, and the proteins displaying enzyme activity were excised. The proteins were digested in-gel with sequencing-grade trypsin, and the peptides were eluted and separated by in-line C18 chromatography, as previously described [14]. In brief, the amino acid sequences of the peptide ions were obtained with an LTQ Orbi-trap mass spectrometer (ThermoFinnigan, San Jose, CA, USA). The b- and y-ion spectra were searched against a database of *R. aeria* F0474, supplemented with decoy proteins as well as the three *Rothia* subtilisin genes published previously [14]. The filter settings selected were X-corr values > 2.2 and 3.5 for Z = 2, and 3, respectively. The deltaCn and peptide probability settings were >0.1 and <0.01, resp. 

Statistical analysis—The in vivo digestion data were analyzed using SPSS 17.0 software and computed with GraphPad Prism 8. The non-parametric Mann–Whitney test was used to test for statistical significance between groups. The data are represented as the average ± standard error of the mean (SEM), and a value of *p* < 0.05 was considered to be statistically significant.

## 3. Results

### 3.1. In Vivo Digestion by life Rothia Bacteria

Mice were fed chow with and without *R. aeria* bacteria, and digestion of the gluten was monitored in samples removed from the digestive tract. During preparation of the chow/*R. aeria* mixture, gliadin remained intact (Appendix A, lanes 2 and 5). In mice fed the chow/*R. aeria* mixture, after 2 h of digestion, most of the chyme was present in the stomach, as evidenced by total protein determination (Figure 1A), SDS page (Figure 1B; left panel) and anti-gliadin immunoblotting (Figure 1B, right panel) of samples harvested from the stomach, duodenum, jejunum, and ileum. 

The in vivo digestion experiments were carried out with 9 female mice per group. After an overnight fast, the control and experimental groups were fed chow without and with *R. aeria* for 1 h, respectively, and euthanized 2 h thereafter. All mice ingested the chow within 1 h, and therefore, the effective digestion time was 2 h. Gliadin levels in the stomach samples from mice fed with and without *R. aeria* were determined from the densitometric analysis of the major gliadin bands at 37 and 50 kDa in sample aliquots normalized to for total protein content (Figure 2A). The gliadin amount significantly decreased in the Ra (+) group compared to Ra (−) group (Figure 2B).

To investigate the survival of immunogenic epitopes in the harvested stomach samples, an ELISA test was carried out using R5 monoclonal antibody, specifically recognizing the QQPFP, QQQFP, and LQPFP immunogenic epitopes [13,17]. The average values for the mice fed with chow supplemented with *R. aeria* was 16.8 ± 0.15 ppm (OD 0.63) compared to 25.1 ± 0.23 ppm (OD 0.88) for the control group. This reduction of the gliadin immunogenic epitopes as detected by ELISA is therefore in accord with the western blot results, showing a statistically significant reduction of 32.6% as compared to the control mice (Figure 2C). 

### 3.2. In Vitro Gluten Digestion by Dead Rothia Bacteria 

In view of the concern that some probiotics, even if they represent harmless colonizers, might favor opportunistic infections in immunocompromised hosts, we investigated if the cell-associated enzyme activities were maintained after abolishing the viability of *Rothia* bacteria. To achieve cell killing with an agent that can be removed after treatment, *R. mucilaginosa* ATCC 25296 and *R. aeria* (WSA-8) bacteria were incubated in 70% ethanol or in saliva ion buffer (SIB, control). After 30 min of incubation, cell viabilities were determined by plating aliquots of the ethanol or SIB-treated cells in triplicate on agar (Figure 3A). Enzyme activities in the ethanol or SIB-incubated cell suspensions were evaluated qualitatively by zymography (Figure 3B), and quantitatively with the tripeptide substrate Z-YPQ-pNA (Figure 3C). The maximum rates of Z-YPQ-pNA hydrolysis are depicted in Figure 3D. Results showed complete loss of cell viability for both strains after incubation with 70% ethanol, but not with SIB (Figure 3A). Remarkably, enzyme activities were maintained, as evidenced from clear bands in the zymogram of the ethanol treated and untreated samples (Figure 3B). The tripeptide hydrolysis rates depicted in Figure 3C,D, show that on average less than 10% of the enzyme activity was lost following 70% ethanol treatment. Overall, the method was an effective, food security compatible approach to abolish bacterial viability without impairing the desired gluten-degrading enzyme activities. 

### 3.3. Isolation and Identification of the Gluten-Degrading Enzyme from R. aeria

To identify the gluten-degrading enzyme from *R. aeria*, a large SDS gel (16 × 20 cm) was used for optimal protein separation. A low percentage gel (6%) facilitated separation of proteins with molecular weights > 50 kDa. From our previous study it was known that the *R. aeria* enzyme migrated at approximately 75 kDa [15]. As an externally added substrate, casein instead of gliadin was added. Casein has a better solubility than gliadin and can be used as alternative substrate to reveal the gliadin-degrading enzymes in zymogram gels, giving a better contrast than gliadin [14,18].

The zymogram provided evidence for a single band migrating at ~75 kDa with enzyme activity, and the matching non-denaturing Coomassie-stained gel showed a major band migrating at the same position (Figure 4A,B). As a control, proteins were also analyzed on a denaturing gel, showing the full spectrum of proteins migrating into the gel under these conditions (Figure 4C). The bands labeled 1–4 in Figure 4A,B were excised and subjected to mass spectrometric analysis. They were searched against a *R. aeria* database, comprising protein entries derived from the whole genome of *R. aeria* as well as protein entries from *R. mucilaginosa* and *R. dentocariosa* subtilisin genes. 

As expected, more than one protein was identified in each band. In total 3, 4, 23, and 23 proteins were identified by >2 unique peptides in bands 1, 2, 3, and 4, respectively. The most prominent protein in all four samples, identified with high confidence, was peptidase S8, KGJ00122.1. It was represented by 32, 30, 33, and 41 unique peptides, in the four excised bands, resp. A blast search of KGJ00122.1 revealed that it is 99% homologous to a *R. aeria* protein annotated as glycerol-3P-ABC transporter in NCBI (BAV86562.1). Sequence analysis of BAV86562.1 revealed that it is a subtilisin family member, because it contains the cd07474 domain that is characteristic of the peptidase S8 family domain. The sequence and the D, S, and H amino acids of the catalytic triad present in this peptidase domain are shown in Appendix A. 

## 4. Discussions 

While oral bacteria are present in high amounts and in a wide variety in the oral cavity, being swallowed in large numbers daily, their roles beyond the oral cavity are not well understood. In the first part of the present work, we demonstrated that *Rothia* species can degrade dietary proteins, specifically gluten, in vitro, as well as in vivo. This suggests that this species, and perhaps some other oral microbes as well, may play a role in the initiation of the digestion of these protein components of foods. 

There is a great interest among CeD patients in a medical therapy that can ameliorate gluten-induced adverse effects that can also occur with trace amounts that celiacs may ingest with a supposed gluten free diet (GFD). A pharmacological therapy should eliminate the effect of milligram quantities up to a few grams of gluten in a largely GFD that avoids overt gluten sources, such as bread, pizza, pasta, or cookies. Most CeD patients agree that such a diet would be much more sustainable, since the required strictly gluten free diet (less than 20 ppm of gluten in all foods, i.e., less than 20 mg/kg) is a great challenge in everyday life [19]. Thus, in a double blind clinical study, CeD patients in remission who were challenged with 50 mg gluten daily developed an average 20% decrease in villous height/crypt depth vs. placebo within 90 days [20], indicating that such minor amounts of gluten can cause chronic mucosal damage. In addition, a high sensitivity to minor amounts of gluten may underly refractory celiac disease type 1. 

A major therapeutic approach to such supportive therapy is the use of bacterial of grain-derived gluten-degrading enzymes, first spearheaded by the group of Khosla [4,21,22,23,24,25,26] while commercially available “glutenases” have no or little proven efficacy to effectively degrade antigenic gluten peptides [26,27]. *R. aeria* is a gram-positive bacterium belonging to the *Rothia* genus and a natural colonizer of the oral cavity, similar to *R. mucilaginosa* and *R. dentocariosa. R. aeria* exhibits a superior gliadin degrading activity compared to *R. mucilaginosa* [15]. The finding suggests that *Rothia* species, in particular *R. aeria*, as nonpathogenic human colonizers, may be considered as promising new candidates for enzyme therapeutics in CeD. 

Gliadin degradation is dependent on the amount of bacteria, the amount of gluten protein, the food matrix, pH, and incubation time. With regard to the bacterial count, we found that *R. aeria* exerted gliadin degradation in a dose dependent manner. Based on in vitro studies employing various ratios of bacteria and gliadin, a dose of 1.6 × 10^7^ CFU/g of chow was selected for the in vivo study. This dose is not unrealistic, considering that probiotic applications can contain as high as 10^11^ up to 10^12^ CFU per application [28]. Given the body weight of mouse of about 20 g, the dose calculated for a 70 kg human would be (1.6 × 10^7^ × 20 × 3500 = ~10^12^). The time between the end of the 1 h chow feed and sacrifice of the mice was 2 h. The selected time span was based on a report that assessed the gastrointestinal transit time in mice with 99mTc-DTPA-labeled activated charcoal. Within 2–3 h the labelled material was detected in the stomach and the entire small intestine [29]. Therefore, 2 h digestion including a 1 h complete ingestion of the food pellet was selected for the in vivo experiments. 

Effective gastric digestion is essential before immunogenic gluten peptides reach the proximal small intestine where they can elicit celiac disease. Therefore, we focused on gastric gliadin digestion. The in vivo experiment with nine mice per group showed that *R. aeria* bacteria significantly reduced the amount of intact gliadin molecules in the stomach before they reached the duodenum. Importantly, this reduction was paralleled by a significant disappearance of major immunogenic gliadin epitopes as assessed with the R5 gliadin ELISA. Rizzello et al. have demonstrated the efficiency of wheat dough fermentation with Lactobacilli and fungal proteases using a similar approach in vitro [30]. 

Gluten-degrading microbes, especially if they are natural colonizers of the human body, are of high interest for exploitation as treatments of CeD, since they carry the potential to be explored in the form of probiotics [26]. Highly effective prolyl- and glutamine-endopeptidases for degradation of immunogenic gluten epitopes are produced by various microorganisms. However, some of the discovered gluten-degrading microbes are opportunistic pathogens and/or demonstrate associations with human diseases. Thus, several gluten-degrading natural colonizers of the human body are not necessarily harmless. Examples are *Capnocytophaga sputigena*, *Neisseria mucosa*, or *Pseudomonas* species, which rapidly cleave immunogenic gluten domains, yet have been implicated in periodontal [31,32,33] and endodontic infections [34], or more general organ pathology [35]. The oral cavity and the gut are usually colonized with balanced quantities of such opportunistic pathogens. The opportunistic/pathogenic features limit or eliminate their potential utilization as probiotic agents; but do not exclude the use of the purified enzymes. 

In this vein, safety is a major concern when live bacteria are considered as probiotic for CeD. The search for an effective and non-toxic probiotic for the treatment of CeD has become a topic of high importance for the celiac community and health care providers [4,22,26,36]. Food-grade proteases capable of detoxifying moderate quantities of dietary gluten could help mitigate this problem [24,26]. In search for alternatives, natural human gluten-degrading microbial colonizers, in a live form when harmless, or neutralized with ethanol treatment, could potentially offer the basis for the development of new and low-cost therapeutics for CeD. In the present study, we developed a method to abolish toxicity associated with the live cells without affecting gluten-degrading activities. The bacterial treatment with 70% ethanol killed the live bacteria, retained gliadin degrading activity, and the added ethanol can be fully removed by lyophilization. While ethanol treatment will not neutralize cell-associated lipopolysaccharide, or certain exotoxins or endotoxins, it abolishes cell viability, colonization capacity, and potential of active transfer of virulence factors and antibiotic resistance genes to other bacteria. Our study also shows that a range of antibiotics (Appendix A) effectively abolished the viability of four gluten-degrading microbial strains. Thus, there are several strategies to safely apply *Rothia* bacteria, e.g., in an ethanol-killed formulation, to patients with CeD, and/or to treat *Rothia* infections effectively with antibiotics, should any infection occur.

We previously identified the gliadin-degrading enzyme from *R. mucilaginosa* [15]. In the present study, we identified the *R. aeria* enzyme a subtilisin-like serine protease belonging to the S8 peptidase family. These subtilisins can rapidly hydrolyze gliadin immunogenic epitopes, efficiently degrade the immunogenic gliadin-derived 33-mer peptide, and can be considered promising new candidates for enzyme therapeutics in CeD [26]. The *R. aeria* protein identified in the gliadin degrading protein band matched with BAV86562.1. This protein was seemingly incorrectly annotated as a glycerol-3-phosphate-ABC transporter in the NCBI data base, since it lacks the structural features of an ABC transporter. ABC transporters are characterized by two transmembrane domains and two nucleotide-binding domains, containing the signature sequence LSGGQ, which is directly involved in nucleotide binding [37]. BAV86562.1 does not contain these domains and the LSGGQ sequence. In contrast, it does contain the D-H-S (Asp-His-Ser) catalytic amino acid triad with flanking regions characteristic for the subtilisin family of proteases. Thus, BAV86562.1 expressed by *R. aeria* is a S8 subtilisin peptidase and not an ABC transporter. 

Peptidase S8 belongs to the subtilases superfamily, a group of enzymes which were first isolated from *Bacillus subtilis*, a species inhabiting, e.g., soils [38]. Since *Bacillus* species are also present in the oral cavity, like *Rothia* species [8], we compared the *R. aeria* and *B. subtilis* gliadin-degrading activities. *R. aeria* was found to have a higher activity than *B. subtilis*. More importantly, *R. aeria*, but not *B. subtilis*, exerted appreciable activity at an acidic pH, i.e., gastric pH 3.0, that is found 1 h after food ingestion [39]. This clearly qualifies the *R. aeria* subtilisin that we identified in this study to serve as an enzyme source that can degrade immunogenic gluten epitopes in the stomach, before they reach the duodenum to elicit intestinal inflammation in patients with CeD. Since the *R. aeria* enzyme can be cloned and expressed in high doses, and since it may be suitable for further optimization towards even higher gluten degrading efficacy in the stomach, clinical applications should be explored. 

## Figures and Tables

**Figure 1 nutrients-12-03724-f001:**
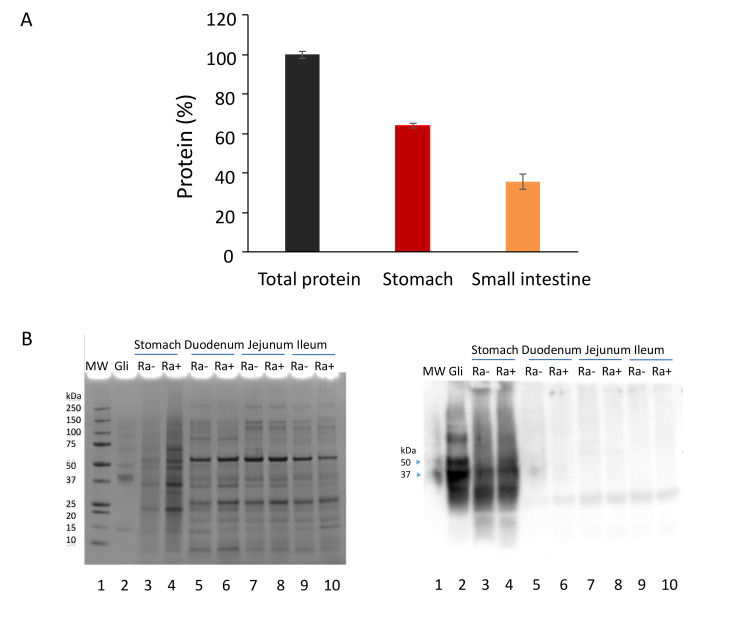
Analysis of gastric and small intestinal contents. (**A**) Total protein content of the contents in the stomach (red) and small intestine (orange); n = 4 per organ; protein was determined by the BCA method multiplied by sample volume and divided by the total protein in both the stomach and intestine. (**B**) Analysis of gliadin in gastro-intestinal samples by SDS-PAGE (left panel) and anti-gliadin immunoblotting (right panel). Aliquots (63 µg protein) of the gastro-intestinal samples with *R. aeria* (Ra+) and without *R. aeria* (Ra−) bacteria were loaded on two 4–12% SDS-PAGE gels. After electrophoresis the gels were either stained with Coomassie brilliant blue (left) or processed by immunoblotting with an anti-gliadin antibody (right). Lanes 1: Protein standard, 5 µL; 2: Gliadin control (Gli, 25 µg); 3: stomach Ra (−);4: stomach Ra (+);5: duodenum Ra (−);6: duodenum Ra (+); 7: Jejunum Ra (−);8: jejunum Ra (+); 9: ileum Ra (−); 10: ileum Ra (+). Arrows: gliadin bands at 37 and 50 kDa.

**Figure 2 nutrients-12-03724-f002:**
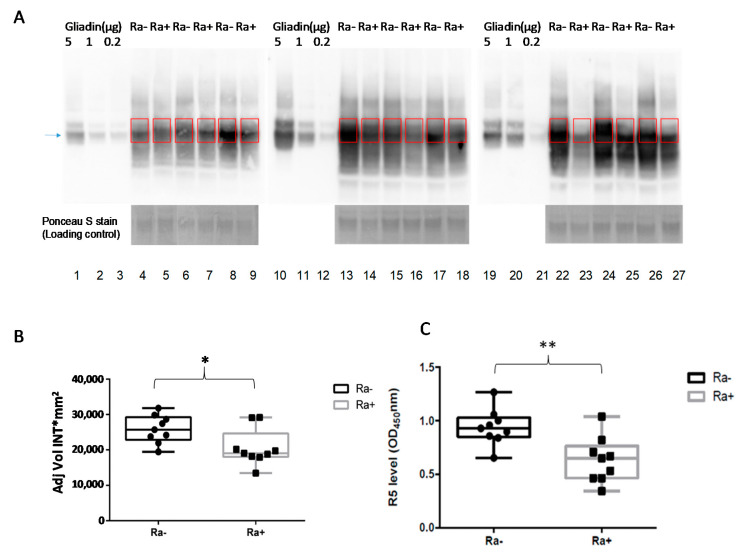
Gliadin degradation in vivo by *Rothia* bacteria. (**A**) Two groups of mice (M) were fed chow without *R. aeria* and chow with *R. aeria*. Aliquots (160 µg protein) of the gastric samples with (Ra+) and without (Ra−) bacteria (n = 9/group) were subjected to immunoblotting with an anti-gliadin antibody. Lane 1, 2, 3; 10, 11, 12; 19, 20, 21: Gliadin control (5, 1 and 0.2 µg), respectively; Lanes 4, 6, 8, 13, 15, 17, 22, 24, 26: the nine mice fed chow without *R. aeria* bacteria Ra (−); Lanes 5, 7, 9, 14, 16, 18, 23, 25, 27: the nine mice fed chow with *R. aeria* bacteria Ra (+). The blue (arrow) pointed to the gliadin bands at 37 kDa. (**B**) Densitometric analysis of the major gliadin protein bands (shown in **A**); data points represent the average of the remaining gliadin amount (AU). The data represent three independent experiments and the error bars ± SEM. * *p* < 0.05. (**C**), R5 epitope levels, expressed in OD450nm, in stomach samples from mice fed chow supplemented with *R. aeria* (Ra+) and without *R. aeria* (Ra−). The data are representative of three independent experiments and the error bars represent the average ± SEM. ** *p* < 0.01.

**Figure 3 nutrients-12-03724-f003:**
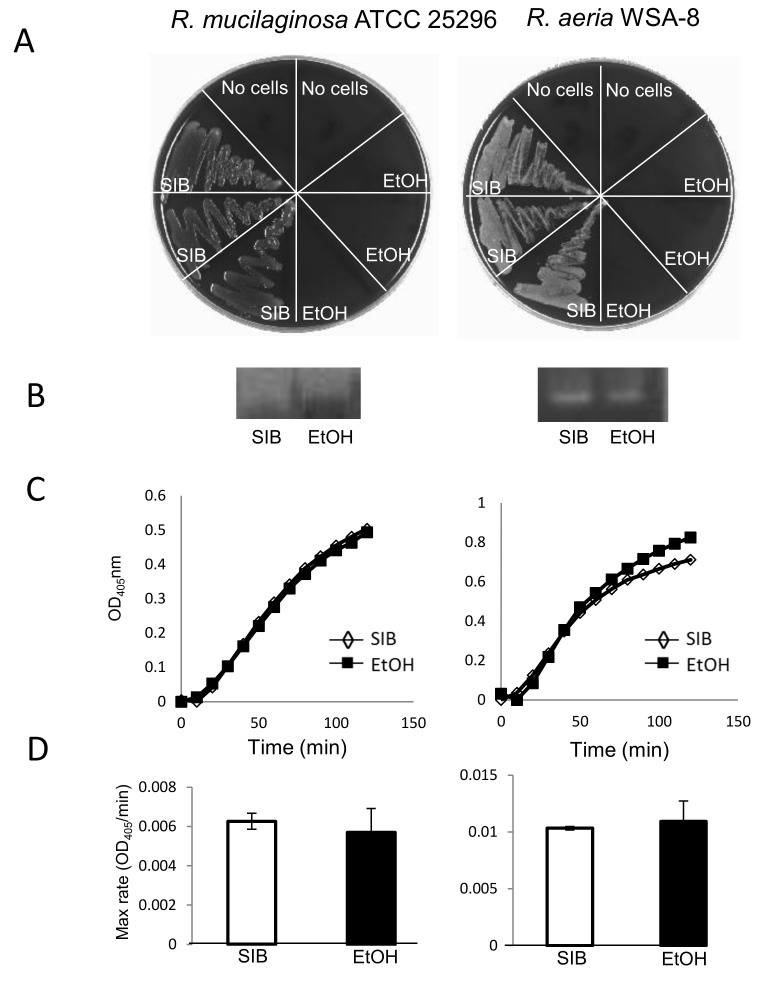
Gliadin epitope degradation by *R. mucilaginosa* and *R. aeria* bacteria incubated in SIB or ethanol. The bacterial cells were suspended in salivary ion buffer (SIB, pH 7.0) or 70% ethanol (EtOH) to a final suspension cell density at OD_620_ of 1.2, and incubated at room temperature for 30 min. After lyophilization and reconstitution, cell viability, and enzyme activities were determined. (**A**) Cell viability assessed by plating 5 µL aliquots in triplicate on BA. (**B**) Cell associated enzyme activities determined by gliadin zymography (zoom of enzyme active band only). (**C**) hydrolysis plots of Z-YPQ pNA (*Rothia* strains). (**D**) Average maximum substrate hydrolysis rates derived from data in (**C**). Left, *R. mucilaginosa* ATCC 25296; right, *R. aeria* WSA-8.

**Figure 4 nutrients-12-03724-f004:**
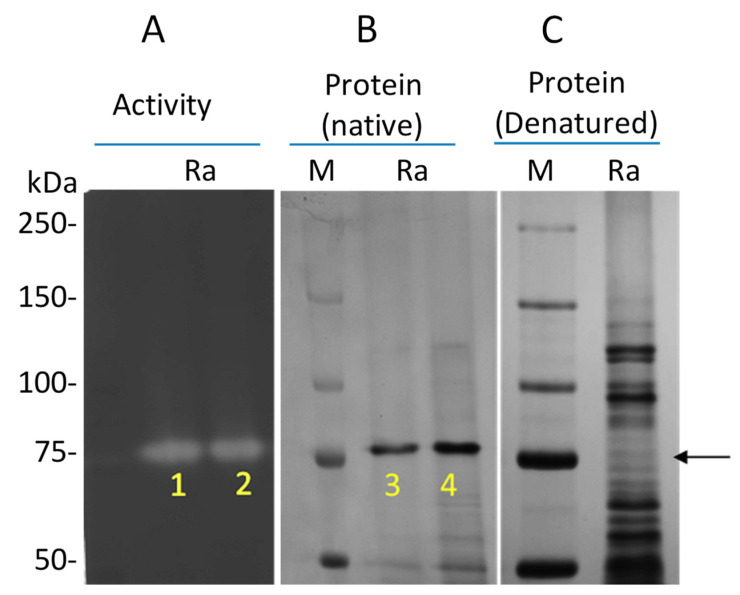
Identification of the gluten degrading enzyme from *R. aeria*. (**A**) Gel part developed as a zymogram gel using externally added casein as the enzyme substrate to visualize bands with enzyme activity (indicated with an arrow). (**B**) Native PAGE gel part stained with Coomassie brilliant blue. (**C**) SDS-PAGE mini gel stained with Coomassie brilliant blue. Bands 1–4 were excised and subjected to mass spectrometric analysis.

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
