# Peer review of "Commensal Bacterium Rothia aeria Degrades and Detoxifies Gluten via a Highly Effective Subtilisin Enzyme"

_nutrients, 2020, doi:10.3390/nu12123724_

Round 1

Reviewer 1 Report

We appreciate the Authors’work; we think that the manuscript is well organized and appropriate.

Author Response

We appreciate the Authors’work; we think that the manuscript is well organized and appropriate.

Response: Thank you. we appreciate your positive assessment.

Reviewer 2 Report

Major comments:

A major claim of the paper is that R. aeria can degrade gluten in vivo. While the data do suggest that R. aeria can degrade gluten (albeit to a limited extent), controls are lacking to demonstrate that this degradation truly occurs in vivo. Specifically, the mouse chow was dissolved with lyophilized bacteria and then dried to reform the dry food pellet. It is entirely possible that all of the gluten degradation occurred ex vivo while generating the bacteria+chow mixture. This control could take the form of an ELISA of an equivalent weight of chow pre- and post- addition of the bacteria. Alternatively or additionally, mice could be fed chow and the gluten degrading bacteria separately rather than premixing the chow and bacteria. While I generally hesitate to request additional experiments given the ongoing pandemic, it is my opinion that the manuscript is not publishable without adding this additional control demonstrating that gluten is not degraded during chow production.

It is not clear in Figure 4C-D exactly what experiments were done. What is the y-axis in panel C? Exactly how were the data processed to yield panel D? These data are important in confirming that even ethanol-killed bacteria can degrade gluten.

The evidence that the gluten-degrading enzyme has been identified is preliminary, and the authors should adjust the wording to acknowledge this. Definitive evidence for this would require demonstrating that genetic knockout of the enzyme abolishes gluten degradation and/or purification followed by biochemical characterization of the enzyme using gluten proteins/peptides as substrates.

Minor comments:

In lines 88-90, it is not clear if additional gliadin was added to the standard mouse chow. If so what was the source of the gliadin? Please revise the wording to make it more clear.

In line 107 and related to Fig 1 and Fig 2, what is the specific anti-gliadin antibody used for Western blotting? Only the manufacturer is listed in the Materials and Methods. The specific antibody should be listed.

The first paragraph of the results could use a sentence or two explaining the rationale and experimental design. Additionally, Figure 1 is a result but is currently only discussed in the materials and methods. Figure 1 should be included at the beginning of the Results section.

The data from Figure 2 and Figure 3A-B should combined into one figure. Additionally, Fig 3A and 3B are different representations of the same data. Only Fig 3B should be included as Fig 3A adds no additional information.

In line with the above, Figure 3C and D show the same data. Only Figure 3D is necessary.

In lines 197-198, the "data not shown" should be included in the supplement.

In Figure 5, what is the justification for using casein zymography instead of gliadin as the substrate?

The discussion is highly speculative and should be more concise and focused on the key conclusions and limitations of the paper.

Author Response

Major comments:

A major claim of the paper is that R. aeria can degrade gluten in vivo. While the data do suggest that R. aeria can degrade gluten (albeit to a limited extent), controls are lacking to demonstrate that this degradation truly occurs in vivo. Specifically, the mouse chow was dissolved with lyophilized bacteria and then dried to reform the dry food pellet. It is entirely possible that all of the gluten degradation occurred ex vivo while generating the bacteria+chow mixture. This control could take the form of an ELISA of an equivalent weight of chow pre- and post- addition of the bacteria. Alternatively, or additionally, mice could be fed chow and the gluten degrading bacteria separately rather than premixing the chow and bacteria. While I generally hesitate to request additional experiments given the ongoing pandemic, it is my opinion that the manuscript is not publishable without adding this additional control demonstrating that gluten is not degraded during chow production.

Response: The chow with and without Ra bacteria was prepared by mixing them as a thick slurry within 1 minute and then immediately drying the mixture in a SpeedVac. During the drying procedure, the slurry freezes up, and the water evaporates under the low pressure.

In Supplemental Figure 1, we checked the auto-degradation of gliadin in the chow/Ra mixture. As evident from lanes 2 and 5, there was no difference in gliadin levels in a chow sample without added Ra and in the chow sample with added Ra at the time =0 time point. This indicates that no digestion of gliadin had occurred during the chow/Ra preparation procedure. This result is now more clearly described in the Results section (p. 5, lines187-192), and also in relation to the interpretation of the in vivo data shown in Figure 2 (p. 5, lines 221-232).

It is not clear in Figure 4C-D exactly what experiments were done. What is the y-axis in panel C? Exactly how were the data processed to yield panel D? These data are important in confirming that even ethanol-killed bacteria can degrade gluten.

Response: We apologize for having omitted the y-axis descriptor; this is now labelled as optical density at 405 nm (OD405), reflecting the pNA release due to substrate cleavage by the gluten degrading bacteria. The slope (change in OD unit per time unit) in C is expressed in Panel D.

The evidence that the gluten-degrading enzyme has been identified is preliminary, and the authors should adjust the wording to acknowledge this. Definitive evidence for this would require demonstrating that genetic knockout of the enzyme abolishes gluten degradation and/or purification followed by biochemical characterization of the enzyme using gluten proteins/peptides as substrates.

Response: Sequential chromatography, combined with zymography and mass spectrometry-based identification, as we presented, are considered pretty hard evidence for the identification of an active enzyme. While we agree that a genetic knockout would be a valuable addition to any identification, we would ask the reviewer and especially the editors to reconsider this demand. Further, it is not unthinkable that Rothia aeria may contain more than one gluten-degrading enzyme, in which case a knockout for the here identified enzyme would still be active towards gluten.

Minor comments:

In lines 88-90, it is not clear if additional gliadin was added to the standard mouse chow. If so what was the source of the gliadin? Please revise the wording to make it more clear.

Response: No additional gliadin but Ra bacteria cells were added to the mouse chow, as stated. The sentence has been rephrased to clarify.

In line 107 and related to Fig 1 and Fig 2, what is the specific anti-gliadin antibody used for Western blotting? Only the manufacturer is listed in the Materials and Methods. The specific antibody should be listed.

Response: We used a validated rabbit anti-gliadin–peroxidase conjugate antibody produced (Sigma A1052-1ML). This information has now been added to the Materials and Methods section (page 3, line 111-112).

The first paragraph of the results could use a sentence or two explaining the rationale and experimental design. Additionally, Figure 1 is a result but is currently only discussed in the materials and methods. Figure 1 should be included at the beginning of the Results section.

Response: We now repositioned Fig.1 in the Results section, and also discuss the data presented there, as suggested.

The data from Figure 2 and Figure 3A-B should combined into one figure. Additionally, Fig 3A and 3B are different representations of the same data. Only Fig 3B should be included as Fig 3A adds no additional information. In line with the above, Figure 3C and D show the same data. Only Figure 3D is necessary.

Response: Thank you for the suggestion. We have modified this accordingly, condensing Fig. 2, Fig. 3B and Fig. 3D into a novel Fig. 2.

In lines 197-198, the "data not shown" should be included in the supplement.

Response:  The information relating to the contents in the stomach and small intestine is now given in Fig. 1A and the text was modified accordingly.

In Figure 5, what is the justification for using casein zymography instead of gliadin as the substrate?

Response: Casein is universal substrate for proteases; it is used to reveal activity of different proteases, including gluten-degrading enzymes; Our previous study (REF#15 and#17) shows that casein can be used as alternative substrate to reveal the same proteolytic activity as gliadin in zymogram gels, with casein giving a better contrast than gliadin. This notion has now been added to the manuscript on p. 9, line 262-264.  Here we aim to better visualize the band of the proteolytic activity, in order to more precisely excise the active band and identify the enzyme by MS.

The discussion is highly speculative and should be more concise and focused on the key conclusions and limitations of the paper.

Response: The Discussion has been modified, especially by omitting a whole paragraph, as marked, to meet the reviewer’s concerns.

Reviewer 3 Report

The objective of this study was to assess gluten-degrading activity of Rothia aeria bacteria in vitro, and to isolate the relative gluten-degrading enzyme.

This is an interesting paper and I have no methodological concerns. The implication of this research are potentially promising for gluten related disorders.

Minor point. For a type error, there is a title of a previous paragraph in the discussion (4.1 in vivo digestion....)

Author Response

The objective of this study was to assess gluten-degrading activity of Rothia aeria bacteria in vitro, and to isolate the relative gluten-degrading enzyme.

This is an interesting paper and I have no methodological concerns. The implication of this research are potentially promising for gluten related disorders.

Minor point. For a type error, there is a title of a previous paragraph in the discussion (4.1 in vivo digestion....)

Response: Thank you, the title has been removed.